# Agnostic Administration of Targeted Anticancer Drugs: Looking for a Balance between Hype and Caution

**DOI:** 10.3390/ijms25074094

**Published:** 2024-04-07

**Authors:** Svetlana N. Aleksakhina, Alexander O. Ivantsov, Evgeny N. Imyanitov

**Affiliations:** 1Department of Tumor Growth Biology, N. N. Petrov Institute of Oncology, 197758 St. Petersburg, Russia; 2Department of Medical Genetics, St. Petersburg Pediatric Medical University, 194100 St. Petersburg, Russia

**Keywords:** agnostic markers, targeted therapy, microsatellite instability, TMB, HRD, kinase inhibitors

## Abstract

Many tumors have well-defined vulnerabilities, thus potentially allowing highly specific and effective treatment. There is a spectrum of actionable genetic alterations which are shared across various tumor types and, therefore, can be targeted by a given drug irrespective of tumor histology. Several agnostic drug-target matches have already been approved for clinical use, e.g., immune therapy for tumors with microsatellite instability (MSI) and/or high tumor mutation burden (TMB), NTRK1-3 and RET inhibitors for cancers carrying rearrangements in these kinases, and dabrafenib plus trametinib for *BRAF* V600E mutated malignancies. Multiple lines of evidence suggest that this histology-independent approach is also reasonable for tumors carrying *ALK* and *ROS1* translocations, biallelic *BRCA1/2* inactivation and/or homologous recombination deficiency (HRD), strong *HER2* amplification/overexpression coupled with the absence of other MAPK pathway-activating mutations, etc. On the other hand, some well-known targets are not agnostic: for example, PD-L1 expression is predictive for the efficacy of PD-L1/PD1 inhibitors only in some but not all cancer types. Unfortunately, the individual probability of finding a druggable target in a given tumor is relatively low, even with the use of comprehensive next-generation sequencing (NGS) assays. Nevertheless, the rapidly growing utilization of NGS will significantly increase the number of patients with highly unusual or exceptionally rare tumor-target combinations. Clinical trials may provide only a framework for treatment attitudes, while the decisions for individual patients usually require case-by-case consideration of the probability of deriving benefit from agnostic versus standard therapy, drug availability, associated costs, and other circumstances. The existing format of data dissemination may not be optimal for agnostic cancer medicine, as conventional scientific journals are understandably biased towards the publication of positive findings and usually discourage the submission of case reports. Despite all the limitations and concerns, histology-independent drug-target matching is certainly feasible and, therefore, will be increasingly utilized in the future.

## 1. Introduction

The first discoveries of oncogenes and suppressor genes in the 1980s revealed that many driver events, for example, *TP53* or *RAS* mutations, are shared between tumors of different types [1]. These findings fueled hopes for the development of “universal” cancer drugs, which are capable of interfering with backbone molecular pathways across a diverse spectrum of malignancies. The invention of mutation-tailored targeted therapies and advent of high-throughput sequencing technologies confirmed that many actionable molecular targets, e.g., *BRAF* V600E substitutions, *ALK/ROS1/RET/NTRK* rearrangements, microsatellite instability (MSI-H), etc., occur at some frequency even in tumors belonging to entirely distinct histological lineages. There is a clear rationale for the use of relevant drugs in an agnostic manner, i.e., in all tumors carrying the appropriate target irrespective of their morphology [2,3,4,5].

Expectedly, the first case reports confirming the success of this approach were met with great optimism. The history of the agnostic targeting of neoplastic growth originates from attempts to utilize tamoxifen, a well-known breast cancer drug directed against the estrogen receptor (ER), for the treatment of ER+ pulmonary lymphangiomyomatosis [6]. Trastuzumab, an anti-HER2 antibody, was approved for the treatment of *HER2*-driven breast cancer in 1998 [7]. The first report demonstrating the agnostic utility of this drug was published just a year later: trastuzumab rendered substantial benefit to a patient suffering from a HER2-overexpressing germ cell tumor [8]. The EGFR inhibitors gefitinib and erlotinib were developed on the assumption that many cancers overexpress this receptor. Clinical trials for these drugs revealed tumor shrinkage only in a small subset of lung cancer patients, and subsequent analysis of tumor tissues obtained from the responders identified unknown drug-sensitizing *EGFR* mutations [9]. The first attempt of the agnostic use of gefitinib is exemplified by its administration to a patient with *EGFR*-mutated renal cell cancer, which resulted in an objective tumor response [10]. In contrast to gefitinib or erlotinib, the development of vemurafenib and dabrafenib relied on the knowledge of the occurrence of *BRAF* V600E mutations across multiple cancer types; therefore, many instances of the agnostic use of these kinase inhibitors were reported immediately after their invention [11,12].

The concept of agnostic drug administration fueled interest in tumor genomic profiling. This approach relies on comprehensive analysis of all known molecular targets in every cancer patient, followed by matching of available drugs to genetic findings. Although the overall promise of reflex DNA and RNA sequencing is beyond any reasonable doubt, some systematic studies brought disappointing results [13,14]. First, the probability of finding the “Achilles heel” in a given tumor is often vanishingly low, as most of the known druggable mutations are exceptionally rare in the agnostic setting [15]. Secondly, even rationally matched treatments sometimes produce limited, if any, benefit [16]. Only a few investigations have assessed real-world attitudes towards agnostic therapy and tumor genomic profiling among medical oncologists and cancer patients [17,18], and there is some contrast between the enthusiasm conveyed by drug manufacturers or genetic service providers and the caution expressed by many practicing physicians or healthcare regulators [15,19,20].

## 2. Approved Agnostic Targets and Drugs

### 2.1. Overview

Several agnostic targets and their matched drugs have been approved by the Food and Drug Administration (FDA) in the past few years [21,22,23]. This list includes microsatellite instability (MSI) and/or deficient mismatch DNA repair (MMR-D) coupled with immune therapy; high TMB (tumor mutation burden), which includes both MSI and some other instances of accumulation of multiple somatic genetic alterations, and is also associated with a response to immune checkpoint inhibitors (ICIs); *NTRK1-3* rearrangements rendering tumor sensitivity to NTRK tyrosine kinase inhibitors (TKIs); *RET* rearrangements calling for the therapeutic use of RET TKI; and *BRAF* V600E mutations associated with tumor responsiveness to combined BRAF and MEK inhibition (Figure 1; Table 1). There are some nuances related to the process of drug approval. For example, both entrectinib and larotrectinib have been subjected to agnostic clinical trials, and, therefore, both of these drugs are permitted for use in *NTRK*-rearranged tumors irrespective of the cancer type. In contrast, only the dabrafenib plus trametinib combination has been successfully evaluated for *BRAF* V600E mutated tumors in the agnostic setting, while similar combinations, i.e., vemurafenib plus cobimetinib, or encorafenib plus binimetinib, have only histology-specific indications for the time being. The European Medicines Agency (EMA) has a more cautious attitude towards tissue-agnostic approvals than the FDA, so the list of agnostic drugs and targets is currently less extensive in Europe than in the USA [23,24,25] (Table 1).

### 2.2. Microsatellite Instability (MSI)/Mismatch Repair Deficiency (MMR-D)

MSI is a relatively straightforward DNA test, which identifies tumors with an accumulation of mutations in microsatellite repeats. These carcinomas have about one to two orders of magnitude more mutations than other malignancies; therefore, they are usually highly antigenic and responsive to immune therapy. Mismatch repair deficiency (MMR-D) is the only known cause of MSI. While MSI detection requires the utilization of molecular genetic tests, the analysis of MMR-D is based on immunohistochemical (IHC) detection of MLH1, MSH2, MSH6, and PMS2 proteins, and, therefore, is perfectly compatible with a standard morphological examination of the tumor tissue. Colorectal carcinomas are the most common tumors with an MSI/MMR-D phenotype. In colorectal cancers, MSI is almost always accompanied by MMR-D, and vice versa; therefore, DNA and IHC tests are interchangeable for this tumor type. This is not true for malignancies with low proliferation rate, in which MMR-D does not result in similarly high accumulation of mutations affecting microsatellites [26,27,28]. Importantly, agnostic clinical trials on pembrolizumab and dostarlimab considered MSI and MMR-D as equivalent tests [29,30]. There is a subset of tumors showing discordant results by MSI and MMR-D, and this discordance is particularly relevant to malignancies arising outside the gastrointestinal tract [31]. Some studies suggest that the use of MMR-D testing alone, i.e., without confirmation of the accumulation of mutations in tumor DNA, may require additional validation, at least in non-colorectal neoplasms [32]. The opposite applies to screening for Lynch syndrome, given that in some rare instances tumors attributed to germline mutations in the *MLH1*, *MSH2*, *MSH6*, or *PMS2* genes may demonstrate MMR-D in the absence of MSI [26,28]. MSI is positioned as an agnostic target, however relevant clinical trials usually included only a few tumor types characterized by high or moderate frequency of MSI/MMR-D occurrence [29,30]. MSI is relatively common in colorectal, gastric, small bowel, and endometrial malignancies, occurs at low frequency in several other cancer types, e.g., biliary tract, pancreatic, or ovarian tumors, but is exceptionally rare in some of the most common oncological diseases, for example, breast or lung carcinomas [33,34,35,36,37]. The detection of MSI involves either the utilization of individually administered IHC or PCR tests, or the use of next-generation sequencing (NGS) panels consisting of several hundred genes [27]. Given that the probability of finding MSI/MMR-D in some cancer types is very small, it is questionable whether indeed all tumor categories deserve MSI/MMR-D testing, unless they arise in subjects with Lynch syndrome.

### 2.3. Tumor Mutation Burden (TMB)

High TMB is an FDA-approved agnostic indication for the use of pembrolizumab. The predictive role of TMB is particularly evident in carcinogen-induced tumors, e.g., lung carcinomas or melanomas. Still, the actual value of TMB determination in these tumor types may be questioned because the history of tumor development is highly indicative of its potential immunogenicity. Indeed, cigarette smoking is a reliable predictor of high TMB in lung cancer, while an excessive number of mutations is observed mainly in melanomas related to sunburns [38,39]. MSI is the most common cause of highly elevated TMB in carcinogen-unrelated tumors. Microsatellite-unstable tumors vary by approximately an order of magnitude with regard to the number of mutations per genome, and, expectedly, the response to immune therapy is as better as more mutations are observed [40,41]. The predictive value of TMB for tumors unrelated to carcinogen exposure or MSI is less proven when considering the commonly accepted threshold of 10 mutations per megabase. Indeed, the agnostic registration trial for pembrolizumab included only a few malignancies whose development was not related to smoking or ultraviolet mutagenic pressure. Strikingly, some relatively common tumor types, for example, breast carcinomas or glioblastomas, do not demonstrate an association between TMB > 10 mutations per megabase and responsiveness to immune therapy [42,43]. There are studies suggesting that increasing the TMB cut-off to >15 mutations/megabase may improve the performance of this marker in an agnostic setting [44]. Also, it is beyond discussion that tumors with ultra-high TMB caused by the deficiency of some DNA repair modules or alterations in DNA polymerase genes are to be prioritized towards the use of immune checkpoint inhibitors [45,46]. Actually, some controversy exists mainly for cancers with moderately increased TMB, with values falling within the range of 10–15 or 10–20 mutations per megabase [44]. Also, technical variations in TMB assays may affect clinical decisions in patients with borderline TMB scores [27,47].

### 2.4. NTRK1-3 Rearrangements

MSI and TMB reflect the integrative characteristics of the tumor genome and, being continuous variables, may have some borderline values. Individual gene rearrangements and mutations are more straightforward targets, as their presence or absence can usually be revealed with a significant level of reliability. *NTRK1-3* fusions were the first categorical variables to receive the recognition of an agnostic target. Two NTRK inhibitors, entrectinib and larotrectinib, received an agnostic approval from the FDA based on the results of the studies, which included a wide spectrum of *NTRK*-rearranged tumors [48,49]. Indirect comparisons of clinical data suggest that larotrectinib may be more efficacious than entrectinib in *NTRK*-driven malignancies [50,51,52]. Although *NTRK1-3* rearrangements are frequently presented as an example of agnostic medicine, their actual utility is limited by some factors. *NTRK1-3* fusions are exceptionally rare in common cancer types; therefore, the feasibility of the use of individual *NTRK1-3* tests is questionable. However, *NTRK1-3* rearrangements are included in the majority of multigene diagnostic panels, so their detection is supported by the rapidly increasing utilization of NGS. While the incidence of NTRK activation in commonly occurring carcinomas affecting adults falls below 1:500–1:1000, *NTRK1-3* translocations are detected at frequency of about 5% in pediatric tumors, sarcomas, and salivary gland carcinomas [53,54]. *NTRK1-3* fusions are particularly common in microsatellite-unstable *RAS/RAF* mutation-negative colorectal carcinomas, while non-colorectal tumors with MSI rarely carry gene translocations [37]. Occasional instances of *NTRK1-3* rearrangements were reported in *KRAS* mutation-negative pancreatic carcinomas and biliary tract cancers [55]. An elevated incidence of *NTRK1-3* translocations was observed in *RET/RAS/RAF* mutation-negative thyroid malignancies and Spitz melanomas [56,57,58]. Current clinical experience of the use of entrectinib and larotrectinib includes no more than a few hundred patients with very diverse tumor types. The data are particularly scarce for *NTRK2* rearrangements, which are significantly less frequent than *NTRK1* or *NTRK3* fusions [48,49,53,54].

### 2.5. RET Rearrangements

RET activation may be caused by amino acid substitutions or gene rearrangements. *RET* mutations are observed in about a half of medullary thyroid carcinomas and occur at some frequency in neuroendocrine tumors. *RET* translocations are particularly characteristic for papillary thyroid carcinomas, being detected in approximately 10–20% of these tumors [59,60,61]. *RET* fusions are also observed in 2–4% of lung adenocarcinomas [62]. Lung cancer is among the most common cancer types in the world, with the majority of subjects presenting with metastatic spread; therefore, this disease is definitively the main source of patients requiring treatment by RET inhibitors [63]. *RET* rearrangements are also common in colorectal tumors with microsatellite instability and Spitz melanomas. Occasional instances of *RET* rearrangements have been described in breast carcinomas, *KRAS* mutation-negative pancreatic cancers, and some other tumor types [37,59,60,64,65,66]. Two RET inhibitors, selpercatinib and pralsetinib, are the standard treatment option for *RET*-driven thyroid carcinomas as well as for *RET*-rearranged non-small cell lung cancers [63,66]. In addition, selpercatinib has already received an agnostic approval for all tumor types carrying *RET* fusion. A selpercatinib pan-cancer clinical trial included patients with 14 tumor types other than lung or thyroid carcinomas; an objective response was observed in 18/41 (44%) subjects [67]. Similar results were reported for pralsetinib, with 13/23 (57%) cases showing complete or partial tumor shrinkage [68].

### 2.6. BRAF V600E

*BRAF* V600E mutations occur in approximately half of patients with skin melanomas, papillary thyroid carcinomas, and Langerhans cell histiocytosis; 5–10% of colorectal cancers and *KRAS* mutation-negative pancreatic malignancies; 1.5–2% of lung carcinomas and biliary tract cancers; and almost all patients with hairy cell leukemia as well as in some other tumor types [12,69]. *BRAF* V600E inhibitors (BRAFi) were initially utilized as single-agents in their first clinical trials. It was quickly revealed that the duration of their effect can be increased by simultaneous inhibition of the downstream target of BRAF, i.e., MEK kinase. Consequently, combined administration of BRAF and MEK inhibitors (MEKi) is the most common approach for the treatment of cancers carrying a *BRAF* V600E substitution. While vemurafenib demonstrated high efficacy against *BRAF*-mutated melanomas, it failed to produce high response rates in colorectal carcinomas [70]. This failure is emphasized by virtually all publications devoted to agnostic therapy as an example of a limitation of the histology-independent consideration of molecular targets [12,16]. In contrast to melanomas, colorectal carcinomas express high amounts of EGFR, and this receptor initiates a collateral signaling cascade in response to BRAF inhibition. In fact, vemurafenib, dabrafenib, or encorafenib do render benefits to colorectal cancer patients, however their use needs to be supplemented by simultaneous administration of anti-EGFR antibodies to avoid feedback activation of the EGFR [71]. Still, there are some controversies which do not have satisfactory explanations. Colorectal, biliary tract, pancreatic, and lung carcinomas are similar with respect to their levels of EGFR expression and EGFR signaling; however, a combined inhibition of EGFR and BRAF is accepted mainly for colorectal malignancies, while the remaining cancer types are treated by BRAF and MEK antagonists. Feasibility of both of the above approaches has been exemplified, at least for gallbladder tumors [12,72,73,74]. Of note, the numerical difference in response rates for BRAFi/MEKi versus anti-EGFR/BRAFi combinations in colorectal cancer is moderate, being 12% for the former but only 26% for the latter [75,76]. The rarity of non-colorectal epithelial *BRAF*-mutated cancers compromises their clinical studies. It may also be easier to arrange a trial involving the combination of BRAF and MEK inhibitors, as both of them are produced by the same manufacturers, while the addition of anti-EGFR antibodies is likely to involve some additional negotiations. An agnostic trial has been carried out for the combination of dabrafenib and trametinib. It included 206 patients with rare varieties of *BRAF*-mutated malignancies. A sufficient number of patients were recruited for anaplastic thyroid cancer, biliary tract cancer, low-grade glioma, high-grade glioma, hairy cell leukemia, and multiple myeloma. A response rate of 33% was observed for high-grade gliomas, while this estimate was equal to or above 50% in the five remaining categories of patients [73]. Similar results were observed in a pan-cancer study utilizing a combination of vemurafenib and cobimetinib [77].

## 3. Other Mutated Oncogenes as Potential Agnostic Targets

Proper clinical evaluation of the agnostic significance of the majority of targets is highly complicated because they are either confined to a single histological entity or exceptionally rare. For example, *EGFR* exon 19 and 21 alterations or *MET* exon 14 skipping mutations occur almost exclusively in lung carcinomas, while *BCR-ABL* rearrangements are characteristic mainly of hematological malignancies [78]. Hence, it is actually impossible to collect a critical number of patients with other than the above tumor types, who have these druggable alterations and receive a matched targeted drug. *ROS1* rearrangements are observed in approximately 1.5% of non-small cell lung cancers, and even for this well-known indication only a few available ROS1 inhibitors have been subjected to a proper registration procedure. The overall number of non-lung *ROS1*-rearranged tumors is vanishingly small, therefore it is highly unlikely that respective clinical experience will be gained in the near future [79,80].

There are several targets and drugs which did not receive formal approval; however, their agnostic significance seems self-explanatory. *ALK* fusions occur in approximately 5% of lung adenocarcinomas. They are also repeatedly found in anaplastic large cell lymphomas, inflammatory myofibroblastic tumors, microsatellite-unstable colorectal carcinomas, *KRAS* mutation-negative pancreatic cancers, thyroid carcinomas, melanomas lacking common oncogene mutations, etc. While the NCI-MATCH trial failed to recruit a significant number of non-lung cancer patients with *ALK* rearrangements [80], there is a critical mass of case reports and small patient series strongly supporting the agnostic utilization of ALK inhibitors [81,82]. ROS1 is highly similar to ALK with regard to biochemical properties and the spectrum of tumor types carrying actionable rearrangements, although its alterations are detected at significantly lower frequencies [79]. The agnostic use of ROS1 inhibitors was considered in the NCI-MATCH investigation; however, only one out of four recruited patients experienced an objective tumor response [80]. Still, several case reports have convincingly demonstrated the utility of ROS1 inhibitors in non-lung cancer types [83,84,85,86,87,88,89,90]. Although case reports are understandably biased towards “positive” findings, they apparently provide a correct picture while justifying a histology-independent use of ALK- and ROS1-targeted drugs.

PIK3CA, a catalytic subunit of PI3K kinase, is a highly attractive target as its activating mutations occur at a reasonable frequency across the most common cancer types, particularly breast, colorectal, and lung carcinomas. PI3K upregulation plays multiple roles in cancer progression. PI3K-driven signaling cascades enhance the survival of transformed cells and thus support their rescue from therapeutic pressure. This property explains the success of combined administration of PI3K inhibitors and endocrine therapy in *PIK3CA*-mutated estrogen receptor-positive breast cancer. Even in this setting, the use of alpelisib was associated mainly with an increase in progression-free survival (PFS), while the gain in overall survival did not cross the threshold for statistical significance [91]. The feasibility of a single-agent administration of PI3K antagonists is disputable. One breast cancer trial demonstrated substantial efficacy of alpelisib monotherapy [92]. In contrast, the use of the PI3K inhibitor taselisib did not render significant benefit to patients with the *PIK3CA* mutation either in a lung cancer study or in an agnostic clinical trial [93,94]. The contribution of *PIK3CA* mutations to cancer phenotype is distinct from other oncogenic drivers. While *EGFR*, *HER2*, *ALK*, *ROS1*, *RET*, *MET*, *NTRK1-3*, *KRAS*, *BRAF*, and other mutations activating the MAPK pathway are usually mutually exclusive and render clear-cut oncogenic addiction, *PIK3CA* alterations trigger a collateral signaling cascade and are very often coincident with MAPK upregulation [3,95]. *KRAS*-mutated tumors are characterized by decreased responsiveness to PI3K pathway downregulation [96]. The PI3K inhibitor copanlisib produced a 16% response rate in the agnostic NCI-MATCH trial; this trial excluded subjects with *KRAS* mutations and some other categories of patients [97].

*KRAS* G12C mutations occur mainly in lung carcinomas obtained from smokers as well as in colorectal carcinomas, although they are occasionally present in other cancer types [48,98]. Clinical trials involving the *KRAS* G12C inhibitors sotorasib and adagrasib demonstrated modest response rates and PFS in lung cancer patients carrying this mutation (28% and 43%, and 5.6 months and 6.5 months, respectively) [99,100]. However, these drugs showed even lower clinical activity when applied towards colorectal cancer (7% and 19%, and 4.0 months and 5.6 months, respectively) [48,101]. Similarly to *BRAF* V600 inhibitors, these drugs appear to be more efficacious for colorectal cancer when administered in combination with anti-EGFR antibodies [101,102]. Responses to sotorasib and adagrasib have also been observed in tumor types other than lung or colorectal; therefore, these drugs appear to have some agnostic potential [48,103].

The FGFR receptor family includes the *FGFR1*, *FGFR2*, *FGFR3*, and *FGFR4* genes. Activating *FGFR1-4* gene rearrangements and point mutations occur at variable frequencies across a diverse spectrum of tumor types. Similarly to other TKIs, FGFR inhibitors demonstrated signs of agnostic antitumor activity [104,105]. The AKT inhibitor capivasertib has demonstrated clinical efficacy in cancers harboring the *AKT1* E17K mutation [106].

## 4. *BRCA1/2* and Homologous Recombination Deficiency (HRD)

The *BRCA1* and *BRCA2* genes were discovered via research on breast–ovarian cancer predisposition. The analysis of BRCA1/2-deficient cell lines and tumors obtained from *BRCA1/2* mutation carriers led to the identification of a therapeutic vulnerability, which was unknown at the time of *BRCA1* and *BRCA2* gene cloning. It has been revealed that BRCA1/2-inactive cells cannot properly repair DNA double-strand breaks due to homologous recombination deficiency (HRD). Consequently, these tumors are highly sensitive to conventional drugs like cisplatin or anthracyclines, which had been invented long before the *BRCA1/2* discovery. The recognition of HRD as a therapeutic target led to the development of PARP inhibitors (PARPis) [107,108]. Germline *BRCA1/2* mutations are particularly characteristic for breast, ovarian, prostate, and pancreatic malignancies. *BRCA1/2* genes are somehow promiscuous with respect to the spectrum of associated tumors; therefore, many other cancer types are slightly enriched by *BRCA1/2* mutation carriers [109]. In addition, the analysis of tumor DNA sometimes results in the finding of somatic mutatios, which occur in subjects with wild-type *BRCA1/2* germ-line DNA [109,110]. For hereditary cancers, the approved PARPi indications for breast, ovarian, pancreatic, and prostate malignancies rely on the fact that, in these tumor types, the germline *BRCA1/2* alteration is almost always accompanied by somatic second-hit inactivation of the remaining allele. In addition, some PARPis may be used for the treatment of phenocopies of the above categories of tumors, e.g., for sporadic prostate or ovarian carcinomas with presumed or established HRD [111].

Substantial limitations are applicable even for currently accepted attitudes towards the use of PARPis. Loss of the remaining *BRCA1/2* allele is observed in the vast majority but still not all hereditary *BRCA1/2*-driven breast and ovarian carcinomas. Expectedly, tumors with preserved *BRCA1/2* heterozygosity are not sensitive to conventional BRCA1/2-specific drugs [111,112,113]. *BRCA1* and *BRCA2* play similar roles in the above cancer types, however their predictive significance for prostate cancer is distinct: indeed, mainly *BRCA2* but not *BRCA1* germline mutations are accompanied by somatic loss of the wild-copy of the gene and consequent sensitivity to PARPis [114,115]. The detection of somatic *BRCA1/2* mutations is sufficient for the administration of PARPis in ovarian and prostate cancers [111]. This indication has been taken from registration trials which permitted the analysis of the tumor tissue only, i.e., without consideration of the status of the germline DNA. Consequently, many subjects included in these trials were in fact hereditary cancer patients. Somatic *BRCA1/2* mutations occur at some frequency in sporadic ovarian and prostate cancers; however, they are not necessarily accompanied by the inactivation of the remaining allele, and, hence, sensitivity to PARPis [113,115,116]. The use of PARPis in prostate cancer relies on the detection of mutations in several homologous recombination genes in addition to *BRCA1* and *BRCA2*. While the predictive role of, for example, mutations in *PALB2* or *RAD51C* appears to be similar to *BRCA1/2*, alterations in some other tested genes, particularly *CHEK2* and *ATM*, are not associated with HRD [110,117,118,119,120].

This situation is getting even more controversial when other than the above tumor types are considered. *BRCA1* and *BRCA2* are included in almost all multigene NGS panels, so their alterations are commonly detected in various categories of malignancies. However, unlike breast and ovarian carcinomas, “non-canonical” malignancies arising in *BRCA1/2* mutation carriers relatively rarely involve inactivation of the remaining gene allele [109,111]. Consequently, agnostic studies utilizing PARPis for the treatment of *BRCA1/2*-mutated cancers demonstrated lower response rates than breast, ovarian, prostate, or pancreatic clinical trials [109,110,111,120]. HRD appears to be a more informative test: it is based not on the detection of mutations in *BRCA1/2* or similar genes, which may or may not render PARPi sensitivity, but on the phenotypic manifestation of deficient DNA double-strand break repair [27]. Although the HRD is very likely to have an agnostic value, this assumption has not been rigorously tested. Clinical trials on the utilization of HRD in the agnostic setting are complicated due to mutual cross-sensitivity and cross-resistance between platinum compounds, anthracyclines, PARPis, and some other drugs [121]. Histology-independent clinical studies usually consider pretreated patients; in contrast, HRD is likely to have a value mainly for the upfront administration of DNA double-strand-break-inducing drugs. HRD is a more sophisticated test than conventional NGS multigene analysis: it relies on pan-genomic scanning of various types of genetic alterations [27,122]. The thresholds for HRD scores appear to vary between different cancer types [109,123]. For example, *BRCA2* mutations are generally associated with lower HRD scores in prostate versus ovarian carcinomas [124]. Overall, treatment decisions based on the results of *BRCA1/2*, HRD, or similar tests are associated with significantly more complexity as compared to relatively straightforward targeting of mutated kinases.

## 5. Oncogene Amplification and Overexpression

While the drug-target relationships discussed above are usually clear-cut, clinical decisions relying on quantitative differences between tumor and norm are more complicated. The increased dosage of oncogenes encoding for tyrosine kinases is frequently observed in various tumor types. Most current NGS-based multigene assays utilize DNA as a template, so these tests are capable of identifying gene amplifications. However, extra gene copies are not necessarily accompanied by overexpression of corresponding RNA transcripts as they may reflect general instability of the tumor genome or accompany amplification-driven activation of neighboring genes located at the same chromosomal locus [125]. Vice versa, gene or protein overexpression does not necessarily involves genomic changes, as it may be caused by epigenetic mechanisms. Furthermore, it is difficult to establish a meaningful threshold for increased gene dosage [4]. For example, different guidelines for the evaluation of *HER2* oncogene activation are applied for breast and stomach cancer, with more strict requirements for the former and seemingly relaxed criteria for the latter [3,126,127]. Comprehensive tumor profiling may help in predicting the drug-target relationships by discriminating between driver and passenger events. Indeed, most clinically relevant amplification or overexpression events affect the MAPK signaling pathway, in which true activating alterations are mutually exclusive. Consequently, the analysis of mutations in *RAS* and *RAF* oncogenes as well as in some relevant receptor kinases may support clinical decisions for tumors with identified extra gene copies [125].

More or less systematic data with regard to agnostic clinical significance of increased gene dosage have been obtained mainly for HER2 receptor tyrosine kinase. *HER2* amplification and overexpression occur at variable frequencies across a wide spectrum of cancer types [126]. Sweeney et al. [125] evaluated the agnostic activity of pertuzumab and trastuzumab in patients with evidence of HER2 activation (amplification and/or overexpression and/or mutation). This MyPathway trial demonstrated encouraging results. Expectedly, the highest clinical benefit was observed for tumors with combined amplification and overexpression (response rate: 39%; PFS: 5.3 months; duration of response: 7.3 months). The efficacy of this therapy was strikingly higher in *KRAS* wild-type versus *KRAS*-mutated tumors. Promising data have also been obtained in the DESTINI-PanTumor02 trial, which utilized trastuzumab deruxtecan for the agnostic treatment of IHC HER2 3+/2+ tumors. There was a strong trend towards improved efficacy of T-DXd in patients with 3+ versus 2+ IHC staining [128]. Overall, the available data suggest that HER2 is indeed an agnostic target, although efforts need to be invested to discriminate between “driver” and “passenger” amplification and/or overexpression.

## 6. PD-L1

PD-L1 is a ligand which interacts with T-cell PD1 receptors and suppresses the antitumor immune response. ICIs interacting with either PD-L1 or PD1 have revolutionized cancer treatment. Soon after the invention of the first ICIs, it was revealed that their effects may be correlated with the level of expression of PD-L1 [129]. Still, there are several controversies and uncertainties related to PD-L1 testing. Most of the currently accepted predictive markers are genetic alterations, i.e., they are categorical variables that are either present or absent in the tumor tissue. Furthermore, mutations are usually stable over time, as their status does not change during the natural history of tumor disease. In contrast, expression-based markers are continuous variables, so clinical decisions are based on the established thresholds [4]. The expression of a given molecule is a more flexible parameter than the presence of mutations. All these limitations are highly relevant to PD-L1 testing. Clinical trials, which demonstrated benefit from ICIs in PD-L1 expressors, utilized different cut-offs. Furthermore, some drugs are tailored to the staining of tumor cells, while others also consider the PD-L1 status of immune cells. The predictive value of PD-L1 may be distinct even for seemingly similar drugs, e.g., pembrolizumab and nivolumab, and these differences are observed even for patients with the same tumor type [130,131,132,133,134]. Importantly, there is no any biological or medical rationale explaining these controversies. Technical variations in PD-L1 testing, including the use of different IHC kits and subjective nature of scoring of PD-L1-positive cells, further compromise this approach [4,130,134]. PD-L1 status may change during chemotherapy, so it is questionable whether PD-L1 analysis of the primary tumor is capable of guiding the administration of ICIs in pretreated patients [135].

The use of PD-L1 testing is complicated even in tumor types with a well-established role for immune therapy. For example, there are many nuances regarding the use of PD-L1 status for the selection of treatment strategy in lung cancer. In fact, PD-L1 determination plays a critical role mainly in the choice of first-line therapy, because tumors with high expression (≥50% tumor cells) can be treated by ICIs without the addition of chemotherapy [131,132,133]. Combined administration of ICIs and cytotoxic drugs, as well as the use of ICIs in pretreated patients, generally do not require determination of PD-L1 status. PD-L1 testing is also not utilized in melanoma treatment, where immune therapy plays a primary role [136]. Despite all these caveats, PD-L1 testing is often included in comprehensive tumor profiling being considered as a potentially agnostic marker [137,138]. Systematic analysis of available data does not confirm the validity of this approach [139]. For the time being, PD-L1 testing may serve as an example of erroneous utilization of tissue-agnostic cancer medicine [4]. While MSI/MMR-D and high TMB have histology-independent significance for the administration of immune therapy, the use of PD-L1 IHC outside the approved clinical indications has to be discouraged [4,139].

## 7. Agnostic Administration of Anticancer Drugs Based on the Results of Comprehensive Tumor Profiling: Clinical Trials

There are several types of clinical trials, which can be categorized as tissue-agnostic investigations. The most straightforward are so-called basket trials: they either utilize well-proven targets, which are established for one or more cancer types, but deserve to be evaluated across a wide spectrum of tumors, or evaluate an agnostic activity of the drug upfront, i.e., without having a prior “success story” in a certain category of cancer patients. Some of these trials have already been referenced in the above sections of this paper. Overall, the majority of basket trials completed so far provided sound evidence for agnostic clinical benefit when dealing with indeed well-defined targets and indeed highly efficient drugs. The appropriate examples include the use of crizotinib in *ALK*-rearranged tumors, or dabrafenib plus trametinib in *BRAF* V600E-mutated malignancies [16]. In addition to histology-agnostic effects, these basket trials have revealed several examples of tumor-specific drug activity [140]. On the other hand, they expectedly often showed no benefit when the definition of target was less strict and/or the efficacy of the applied drug had not been previously demonstrated in at least one tumor type. For example, afatinib has shown clinical efficacy against lung cancers harboring exon 19 deletions, L858R substitutions, or some other *EGFR* activating mutations located in exons 18–21 of the *EGFR* gene [9]. Consequently, the attempt to use it against tumors with *EGFR* exon 13 R521K mutations appears to be poorly justified [141]. Of note, the same study demonstrated a clinical benefit from this drug in a patient with an *EGFR* L858R-mutated salivary gland tumor [141]. Afatinib has shown no benefit in patients with *HER2*-mutated lung carcinomas [142]. Not surprisingly, the use of this drug against the same target in an agnostic setting failed to render histology-independent benefit to cancer patients [143]. Amplifications of *CCND1*, *CCND2*, and *CCND3* are frequently observed across multiple cancer types. Despite initial expectations, they have not been confirmed to be reliable predictive biomarkers for inhibitors of cyclin-dependent kinases (CDKs) in breast cancer studies [144]. Consequently, CDK inhibition has not demonstrated promising results in *CCND*-amplified malignancies [145].

Some trials utilized comprehensive genomic profiling for a certain tumor type. A pancreatic cancer trial demonstrated improved overall survival in patients who received therapy matched to genomic aberrations. Still, the most pronounced responses were observed either in patients with already approved organ-specific drug indications (e.g., olaparib in *BRCA1/2*-driven pancreatic cancer), or in subjects with overtly actionable lesions (*ALK* fusions, *BRAF* V600E mutations, microsatellite instability, etc.) [146]. Similarly, the vast majority of patients benefiting from genomically-tailored therapy in a recently reported breast cancer trial either had a *PIK3CA* mutation and received alpelisib, or had *BRCA1/2* or similar alterations and received olaparib. There was no benefit from matched treatment in women whose tumors carried alterations with the level of actionability below I/II according to the ESMO Scale for Clinical Actionability of Molecular Targets (ESCAT) [113].

Earlier trials invited heavily pretreated patients with any tumor type, and attempted to match therapy both to clearly druggable and to apparently less actionable genetic alterations [147]. In contrast to target-centered or organ-centered trials (see above), in which drug-target matching was agreed at the start of the study, these investigations allowed some flexibility in treatment decisions on the case-by-case basis [13,137,148,149]. Some trials considered only genomic alterations, while others supplemented DNA testing with RNA expression profiling or IHC analysis of selected molecules [136,149,150]. Overall, these studies showed mixed results, as they included highly heterogeneous categories of patients and treatments, and were not easily amenable to quantitative assessment [15]. However, a subgroup analysis of some of these trials has provided encouraging data [151,152].

## 8. Conclusions and Perspectives

The mere feasibility of an agnostic approach has never been subjected to a doubt. However, many nuances related to the use of cancer drugs in a histology-independent manner remain to be investigated and discussed (Table 2). Obviously, the actual value of the molecular targets ranges from almost a panacea to a fiction. Some conflict-of-interest is probably involved in this field, as providers of genetic services are sometimes interested in the overinterpretation of genetic findings in order to increase the attractiveness of tumor genomic profiling. ESMO has developed the ESCAT criteria, which provide a framework for ranking the targets according to their potential druggability [153]. The feasibility of using ESCAT has been confirmed in clinical studies [133]. Still, treatment decisions for individual patients are almost always associated with significant level of complexity; therefore, the majority of cancer centers utilize Molecular Tumor Boards (MTBs) for the personalization of treatment choices [13,137,148,149,154]. MTBs are compatible with general attitudes in cancer medicine, which rely on tight interaction of physicians with complementary expertise. While conventional treatment discussions usually involve surgeons, medical oncologists, and radiologists, MTBs require an additional input from pathologists, molecular geneticists, and translational specialists, i.e., the professionals who are fluent in essential technical and fundamental aspects related to the personalization of targeted therapy. In many instances, the determination of the potential druggability of a given molecular feature looks relatively straightforward; however, it is more difficult to compare the chances of deriving benefit between experimental and standard treatment options. *BRAF* mutations may serve as a relevant example: indeed, *BRAF* V600E-driven malignancies demonstrate significant histology-specific variations in the efficacy of BRAFi/MEKi, and many of these tumors can be efficiently managed by other than BRAFi/MEKi therapies [75,76,77].

There are many examples in modern oncology when tumors have more than one molecular target, and in these cases there are difficult choices with regard to the sequence of the treatments and the feasibility of combined therapy [155]. The best-known examples include sun-induced melanomas with high tumor mutation burden and *BRAF* V600E mutations, smoking-related lung carcinomas with *KRAS* G12C substitutions, and microsatellite-unstable colorectal tumors with kinase-activating genetic alterations, which are potentially sensitive both to immune therapy and to inhibitors of the MAPK pathway [37,98,156]. For example, clinical studies on *BRAF* V600E mutant melanomas revealed that the overall survival is higher in patients who received BRAFi/MEKi after disease progression during immune therapy when compared to subjects treated with BRAFi/MEKi upfront and ICIs as a second-line [157,158]. Furthermore, combined administration of BRAFi/MEKi and ICIs seems feasible in patients who developed resistance to these drugs previously given as sequential treatment [159]. It is of interest whether the principles, which have been obtained for melanoma, are applicable towards other cancer types.

Agnostic clinical trials are highly demonstrative as a proof-of-concept, but obviously cannot cover the entire spectrum of clinical situations. By definition, even the most comprehensive studies are characterized by uneven histological distribution of target-containing tumors, with many cancer categories often represented by one patient each. Furthermore, these investigations often do not account for many important nuances: for example, subjects with brain metastases are usually considered as an especial entity in tumor-specific trials, however this issue has not received a proper attention in the pan-cancer evaluation of targeted drugs [16,29,30,42,44,48,49,67,68,73,77,80,94,97,103,104,105,125,128]. The accumulation of a real-world experience is critical for the progress of agnostic cancer medicine. The existing format of dissemination of treatment experience relies mainly on publications in peer-reviewed journals. However, medical journals usually discourage presentation of case observations, and the bias towards positive findings is a significant problem for research reports. It is probable that personalized cancer medicine requires more accessible and more flexible format of data dissemination; however, proper quality assurance and reliability are an issue for many internet-based data sources. For example, significant differences have been observed for mutation frequencies presented in databases and published clinical studies [16].

Despite all these controversies, comprehensive NGS-based genomic and transcriptomic profiling is very likely to become a mandatory procedure for tumor analysis in the future, similarly to a standard morphological examination or routine clinical tests. There are opinion leaders suggesting that the existing organ-based or histology-based approach needs to be replaced by target-based tumor classification [138]. It is necessary to keep in mind that the results of comprehensive molecular profiling cannot be properly interpreted irrespective of the clinical and tissue context, as the same molecular events demonstrate significant variations in druggability across different types of cancer [140]. Although some caveats and limitations exist, the histology-agnostic approach will probably become a prevailing attitude in targeted cancer treatment in the next few years.

## Figures and Tables

**Figure 1 ijms-25-04094-f001:**
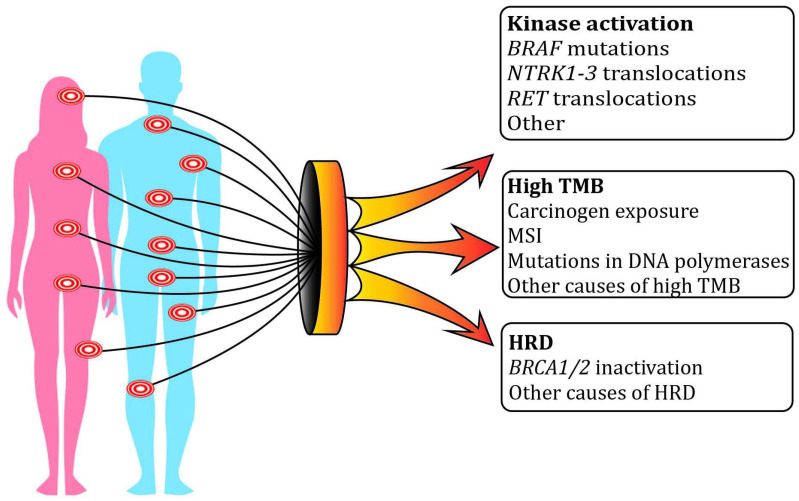
The concept of agnostic therapy. Some tumor vulnerabilities are shared across various cancer types; therefore, their targeting is potentially effective irrespective of the histological origin of a given malignancy. Activating mutations in kinases involved in the MAPK signaling pathway are the best established agnostic targets. High tumor mutation burden, which may be caused by external carcinogens, errors in DNA repair pathways, or alterations in DNA polymerases, is associated with increased tumor antigenicity and, therefore, its sensitivity to immune therapy. Homologous recombination deficiency may be attributed to the inactivation of *BRCA1/2* or some other genes; it renders tumor responsiveness to DNA double-strand-break-inducing drugs.

**Table 1 ijms-25-04094-t001:** Approved agnostic drugs and corresponding clinical indications.

Drug	FDA	EMA
Pembrolizumab	Tumors with MSI/MMR-D or high TMB that have progressed following prior treatment and which have no satisfactory alternative treatment options	Colorectal, endometrial, gastric, small intestine or biliary tumors with MSI/MMR-D, but not a truly agnostic approval
Dostarlimab	Tumors with MMR-D that have progressed during or following prior treatment and who have no satisfactory alternative treatment options	-
Entrectinib, larotrectinib	*NTRK*-rearranged tumors that have progressed following treatment or have no satisfactory alternative treatment options	*NTRK*-rearranged tumors that have no satisfactory alternative treatment options
Selpercatinib	*RET*-rearranged tumors that have progressed following treatment or have no satisfactory alternative treatment options	-
Dabrafenib plus trametinib	Tumors with *BRAF* V600E mutations that have progressed following prior treatment and which have no satisfactory alternative treatment options	-

Data are taken from https://www.accessdata.fda.gov/ (last accessed on 17 February 2024) for FDA and from https://www.ema.europa.eu/en/medicines/human/EPAR/ for EMA (last assessed on 17 February 2024); explanations underlying gene names are given in Appendix A.

**Table 2 ijms-25-04094-t002:** Examples of established, controversial, and mistaken agnostic targets in cancer therapy.

Target	Current Status	Outstanding Issues
Approved target-drug matches		
MSI/MMR-D	Approved indication for pembrolizumab and dostarlimab (previously treated tumors only)	Feasibility of the first-line administration of ICIs; interchangeability of different ICIs; interchangeability of MSI and MMR-D testing in non-gastrointestinal tumors
High TMB	Approved indication for pembrolizumab (previously treated tumors only)	Feasibility of the first-line administration of ICIs for tumors with ultra-high TMB; interchangeability of different ICIs; proper choice of the TMB threshold
*NTRK1-3* rearrangements	Approved indication for entrectinib and larotrectinib	
*RET* rearrangements	Approved indication for selpercatinib; similar tumor-agnostic efficacy has already been demonstrated for pralsetinib	Feasibility of agnostic use of RET inhibitors for tumors with activating *RET* mutations
*BRAF* V600E	Approved indication for dabrafenib plus trametinib (previously treated tumors only); similar tumor-agnostic efficacy has already been demonstrated for vemurafenib plus cobimetinib	Feasibility of the combined use of BRAF inhibitors and anti-EGFR antibodies in other than non-colorectal tumors of epithelial origin
Highly convincing target-drug matches		
*ALK* or *ROS1* rearrangements	Although definite clinical trials have not been carried out, the available evidence indicate that the principles obtained for *NTRK1-3* and *RET* rearrangements are applicable for *ALK* and *ROS1* fusions	
*HER2* overexpression due to gene amplification	Multiple lines of evidence suggest benefit from anti-HER2 antibodies; exclusion of other activating events in MAPK pathway, particularly RAS hot-spot mutations, is a strong prerequisite for this therapy	The choice between anti-HER2 therapy alone versus combination of anti-HER2 antibodies with chemotherapy
Target-drug matches requiring additional consideration		
*PIK3CA* mutations	Limited efficacy of PI3K inhibitors in agnostic clinical trials	The benefit from PI3K inhibitors may depend on the molecular tumor context, e.g., the status of MAPK cascade; although having limited clinical efficacy as single agents, PI3K inhibitors may exert synergism with other therapeutic compounds
*KRAS* G12C	Moderate efficacy of sotorasib and adagrasib both in tumor-specific and agnostic clinical trials	
*FGFR1-4* rearrangements and mutations	*FGFR* inhibitors have demonstrated signs of pan-tumor activity in agnostic clinical trials, although more data remain to be accumulated	
*BRCA1/2* mutations, mutations in “BRCA-like” genes, HRD	Agnostic clinical studies have been carried out for PARP inhibitors and produced controversial results; preclinical data do not support the agnostic significance of *BRCA1/2* etc. mutations; despite these limitations, some genomic profiling services mistakenly use the analysis of mutations in *BRCA1/2* and related genes as a standalone predictive test	The analysis of somatic “second-hit” mutation in the tumor tissue is a mandatory requirement for the treatment decisions when considering carriers of *BRCA1/2* pathogenic alleles. While some genes (*PALB2*, *RAD51* family, etc.) are equivalent to *BRCA1/2*, several other genes (*ATM*, *CHEK2*, etc.) need to be removed from the drug-matching tumor genomic profiling. HRD is a highly promising agnostic marker deserving evaluation in clinical trials; HRD-tailored clinical studies should focus on treatment-naive but not chemotherapy-pretreated patients. Potential interchangeability of PARP inhibitors, platinum salts, metronomic oral cyclophosphamide, etc. requires consideration
Examples of mistaken target-drug matches		
PD-L1 expression	Neither preclinical nor clinical data support the use of PD-L1 testing in agnostic setting; however, some diagnostic services mistakenly offer PD-L1 IHC as a pan-tumor assay	For the time being, the use of PD-L1 IHC for treatment decisions outside the approved indications has to be strongly discouraged
*CCND1-3* gene amplification	Although the predictive significance of *CCND1-3* gene status has not been proven, some genomic profiling services mistakenly interpret *CCND1-3* gene amplification as a rationale for the use of CDK inhibitors	

Explanations underlying gene names are given in Appendix A.

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
