# Peer review of "Agnostic Administration of Targeted Anticancer Drugs: Looking for a Balance between Hype and Caution"

_ijms, 2024, doi:10.3390/ijms25074094_

Round 1

Reviewer 1 Report

Comments and Suggestions for Authors

The review describes existing approved and potential agnostic targets and drugs, and the ways of them realization. The authors have a good understanding of the research topic. The bibliography contains recent sources and publications. It should be noted that the presented review is novel and differs from existing ones (for example, Pharmaceuticals (Basel), 2023, doi: 10.3390/ph16040614; Adv Ther. 2023, doi: 10.1007/s12325-022-02362-4; Expert Review of Molecular Diagnostics , 23, doi: 10.1080/14737159.2023.2245752), which indicates the authors’ involvement in this topic and defending their own point of view on the existing problem. Therefore, I consider it possible to accept this review for publication in present form.

Author Response

Thank you very much for positive evaluation of our work!

Reviewer 2 Report

Comments and Suggestions for Authors

The review entitled “Agnostic administration of targeted anticancer drugs: looking for a balance between hype and caution” by Aleksakhina et al. summarizes the current FDA-approved agnostic anti-cancer drugs available for patients with advanced or metastatic cancers highlighting both enthusiasm and caution towards this novel type of precision oncology therapy.

Overall, the authors are presenting a well-structured and concise review in good English language. According to the title, the article aims to ponder controversies being associated with agnostic drug treatment, however, the arguments in favor (hype) and against (caution) are quite hidden in the text. Giving all information in a factual, plain text, the reading experience is rather dry and it is hard to view every information.

Specific comments:

1. Please, add some subheadings per paragraph (2.-7.) and summarize key points in a table at the end of the article, for instance, present each molecular/genetic alteration -approved and potential targets - and the advantages and disadvantages for their consideration as tumor-agnostic drug target.

2.     Page 2: Could the authors please provide any information of which tissue-agnostic drugs are currently being EMA approved. Please, add one sentence in line 96.

3.     Page 3, Figure 1: Figures and figure captions usually should be descriptive enough to be understood without having to refer to the main text. Hence, the illustration representing the concept of agnostic therapies requires more explanation. Please extend the information given in the figure legend or add annotations to the image.

4.     Abbreviations are not sufficiently explained. Hence, phrases should be spelled out in full and abbreviations should be provided in parentheses the first time a phrase is mentioned. Thereafter, abbreviations only can be used.

For example:

-       Line 13: “NTRK1-3”; please write “Neurotrophe Tyrosin-Rezeptor Kinase 1-3 (NTRK1-3)”.

-       Line 14: “RET”; write “Rearranged during transfection (RET)”.

-       etc.

5.     Page 11, line 528: Since no supplementary material was attached to this review, please comment “Not applicable” after the colon.

Author Response

Comment: According to the title, the article aims to ponder controversies being associated with agnostic drug treatment, however, the arguments in favor (hype) and against (caution) are quite hidden in the text. Giving all information in a factual, plain text, the reading experience is rather dry and it is hard to view every information.

Response: We have added Tables 1 and 2 in order to deliver the information in a more structured way.

Comment: Please, add some subheadings per paragraph (2.-7.) and summarize key points in a table at the end of the article, for instance, present each molecular/genetic alteration -approved and potential targets - and the advantages and disadvantages for their consideration as tumor-agnostic drug target.

Response: We have incorporated subheading for the Section 2, which is the most extensive section of the manuscript. Please allow us to abstain from inserting subheadings in the remaining sections: they are all relatively short, and their excessive fragmentation may affect the perception of the message.  We have also inserted Table 2, which summarizes current status of the most studied agnostic targets and points at some concerns related to their use. 

Comment: Page 2: Could the authors please provide any information of which tissue-agnostic drugs are currently being EMA approved. Please, add one sentence in line 96.

Response: We have added Table 1, which provides information on the FDA and EMA approval status of agnostic drugs.

Comment: Page 3, Figure 1: Figures and figure captions usually should be descriptive enough to be understood without having to refer to the main text. Hence, the illustration representing the concept of agnostic therapies requires more explanation. Please extend the information given in the figure legend or add annotations to the image.

Response: We have added explanations to this Figure in the figure legend.

Comment: Abbreviations are not sufficiently explained. Hence, phrases should be spelled out in full and abbreviations should be provided in parentheses the first time a phrase is mentioned. Thereafter, abbreviations only can be used.

For example:

-       Line 13: “NTRK1-3”; please write “Neurotrophe Tyrosin-Rezeptor Kinase 1-3 (NTRK1-3)”.

-       Line 14: “RET”; write “Rearranged during transfection (RET)”.

Response: Abbreviations used for gene names often reflect rather the historical context of the discovery of these genes than their function. In addition, the text may look very cumbersome if the explanation will be given after first mention of each gene name. Hence, we decided to provide all current and previously utilized gene names in Supplementary Table S1.

Comment: Page 11, line 528: Since no supplementary material was attached to this review, please comment “Not applicable” after the colon.

Response: We have added Supplementary Table S1 explaining gene names.

Reviewer 3 Report

Comments and Suggestions for Authors

Svetlana N. Aleksakhina and coauthors addressed the important topic of agnostic anticancer drug use in the clinical setting.

1)      The first point, I would like to stress, regards the statement, line 9.  I agree that many tumors harbor well-defined vulnerabilities however, this does not always translate to specific effective treatments, especially when we consider Kinase activation pathways, where the targeting of the same pathway, e.g. BRAF mutation resulted in different activity and efficacy in different tumors as   NSCLC, colon, melanoma, anaplastic thyroid carcinoma, biliary tract, low and high-grade glioma, hairy cell leukemia, multiple myeloma, and the differences are more evident when we consider PFS and OS.

2)      Another aspect is the paucity of data on rare mutations, and fusion rearrangements (ROS1, ALK, MET, NTRK1-3) on rare tumors) making it difficult to build up enough evidence to support the clinical choice. The line and type of previous treatment, clinical patient conditions, metastatic sites, and, tumor burden, could also have a high impact on the efficacy, and for that reason, adequate planned basket trials, with adequate selection criteria, follow-up, and translational  data, are at the moment the only way to acquire scientific evidence adequate for regulatory drug agencies.

3)      I agree on  considerations on the PD-L1 testing, scoring, and use for immunotherapy choice.  At the moment data supporting its use are frail and confounding, even when derived from CRTs. I think this topic should be considered together with, TMB, MSI. and MMR-D on  section 2, since it refers to the immunotherapy choice

4)      Lines 504-510 afford the important issue of treatment sequencing when more than one actionable target is present. At least in metastatic BRAF mutated melanoma, the sequencing of combo immunotherapy (Nivo-Ipi) followed by anti-BRAF plus anti-MEK treatment has been proved superior to the reverse in two randomized trials (DreamSeq and Secombit) and the real-world data, derived from the EuMelReg, DeCog registries, reinforce this evidence.

5)       You did highlight the effort made in this field by ESMO with the ESCAT scale construction and implementation, and I strongly suggest its use in clinical practice, as well, to implement the referring to Molecular Tumor Boards for ESCAT below level I and II.

6)      I share your opinion that NGS-based genomic and transcriptomic profiling would become a common or mandatory tumor analysis in the future, however, I think that different  gene analysis pathways should be implemented for different tumors, considering the diversity in hierarchal effects of genomic organization in different histology.

7)      Last consideration, the time and site of biopsy are fundamental, moreover in immunotherapy choice. In the clinic, there is an absolute need for acquiring data on the brain metastasis development risks related to the treatment choice. We know that some effective treatments are aggravated by an increased risk of brain failure compared to others, and we need to improve our knowledge in that field.

8)      The references are adequate and exhaustive and I congratulated the authors for this effort.

Author Response

Comment: The first point, I would like to stress, regards the statement, line 9.  I agree that many tumors harbor well-defined vulnerabilities however, this does not always translate to specific effective treatments, especially when we consider Kinase activation pathways, where the targeting of the same pathway, e.g. BRAF mutation resulted in different activity and efficacy in different tumors as   NSCLC, colon, melanoma, anaplastic thyroid carcinoma, biliary tract, low and high-grade glioma, hairy cell leukemia, multiple myeloma, and the differences are more evident when we consider PFS and OS.

Response: We have softened the first statement in the Abstract by inserting the word “potentially”: “…Many tumors have well-defined vulnerabilities, thus potentially allowing highly specific and effective treatment….”. We have also incorporated the above example on BRAF mutations in the Section 8, where we discuss limitations of the agnostic approach “…BRAF mutations may serve as a relevant example: indeed, BRAF V600E-driven malignancies demonstrate significant histology-specific variations in the efficacy of BRAFi/MEKi, and many of these tumors can be efficiently managed by other than BRAFi/MEKi therapies [75-77]”.

Comment: Another aspect is the paucity of data on rare mutations, and fusion rearrangements (ROS1, ALK, MET, NTRK1-3) on rare tumors) making it difficult to build up enough evidence to support the clinical choice. The line and type of previous treatment, clinical patient conditions, metastatic sites, and, tumor burden, could also have a high impact on the efficacy, and for that reason, adequate planned basket trials, with adequate selection criteria, follow-up, and translational  data, are at the moment the only way to acquire scientific evidence adequate for regulatory drug agencies.

Response: We have now addressed this issue in the Section 8: “By definition, even the most comprehensive studies are characterized by uneven histological distribution of target-containing tumors, with many cancer categories often represented by one patient each. Furthermore, these investigations often do not consider many important nuances: for example, subjects with brain metastases are usually considered as an especial entity in tumor-specific trials, however this issue has not received a proper attention in the pan-cancer evaluation of targeted drugs [16,29,30,42,44,48,49,67,68,73,77,80,94,97,103,104,105,125,128].”

Comment: I agree on  considerations on the PD-L1 testing, scoring, and use for immunotherapy choice.  At the moment data supporting its use are frail and confounding, even when derived from CRTs. I think this topic should be considered together with, TMB, MSI. and MMR-D on  section 2, since it refers to the immunotherapy choice

Response: Please let us disagree. We discuss MSI/MMR-D and TMB in the section devoted to well-established and properly justified agnostic markers. It would be confusing if we move PD-L1 to this section. In order to strengthen the message, we have added to the discussion on PD-L1 the following statement: “…While MSI/MMR-D and high TMB have histology-independent significance for the administration of immune therapy, the use of PD-L1 IHC outside the approved clinical indications has to be discouraged [4,139].”

Comment: Lines 504-510 afford the important issue of treatment sequencing when more than one actionable target is present. At least in metastatic BRAF mutated melanoma, the sequencing of combo immunotherapy (Nivo-Ipi) followed by anti-BRAF plus anti-MEK treatment has been proved superior to the reverse in two randomized trials (DreamSeq and Secombit) and the real-world data, derived from the EuMelReg, DeCog registries, reinforce this evidence.

Response: We have now commented on these studies in the Section 8: There are many examples in modern oncology when tumors have more than one molecular target, and in these cases there are difficult choices with regard to the sequence of the treatments and the feasibility of combined therapy [155]. The best known examples include sun-induced melanomas with high tumor mutation burden and BRAF V600E mutations, smoking-related lung carcinomas with KRAS G12C substitutions, and microsatellite unstable colorectal tumors with kinase-activating genetic alterations, which are potentially sensitive both to immune therapy and to inhibitors of the MAPK pathway [37,98,156]. For example, clinical studies on BRAF V600E mutant melanomas revealed, that the overall survival is higher in patients who received BRAFi/MEKi after the disease progression on immune therapy when compared to subjects treated with BRAFi/MEKi upfront and ICIs in the second-line [157,158]. Furthermore, a combined administration of BRAFi/MEKi and ICIs seems feasible in patients who developed the resistance to these drugs previously given as sequential treatment [159]. It is of interest whether the principles, which have been obtained for melanoma, are applicable towards other cancer types.

Comment: You did highlight the effort made in this field by ESMO with the ESCAT scale construction and implementation, and I strongly suggest its use in clinical practice, as well, to implement the referring to Molecular Tumor Boards for ESCAT below level I and II.

Response: We have now extended the comments on this issue: “…ESMO has developed the ESCAT criteria, which provide a framework for ranking the targets according to their potential druggability [153]. The feasibility of using ESCAT has been confirmed in clinical studies [133]. Still, treatment decisions for individual patients are almost always associated with significant level of complexity, therefore, the majority of cancer centers utilize Molecular Tumor Boards (MTB) for the personalization of treatment choices [13,137,148,149,154]. MTBs are compatible with general attitudes in cancer medicine, which rely on tight interaction of physicians with complementary expertise. While conventional treatment discussions usually involve surgeons, medical oncologists and radiologists, MTBs require an additional input from pathologists, molecular geneticists, translational specialists, i.e., the professionals who are fluent in essential technical and fundamental aspects related to the personalization of targeted therapy.  …”.

Comment:  I share your opinion that NGS-based genomic and transcriptomic profiling would become a common or mandatory tumor analysis in the future, however, I think that different  gene analysis pathways should be implemented for different tumors, considering the diversity in hierarchal effects of genomic organization in different histology.

Response: We have now commented on the importance of accounting for tumor type: “…There are opinion leaders suggesting that the existing organ-based or histology-based approach needs to be replaced by target-based tumor classification [138]. It is necessary to keep in mind that the results of comprehensive molecular profiling cannot be properly interpreted irrespective of the clinical and tissue context, as the same molecular events demonstrate significant variations in druggability across different types of cancer [140]…”.

Comment: Last consideration, the time and site of biopsy are fundamental, moreover in immunotherapy choice. In the clinic, there is an absolute need for acquiring data on the brain metastasis development risks related to the treatment choice. We know that some effective treatments are aggravated by an increased risk of brain failure compared to others, and we need to improve our knowledge in that field.

Response: We have inserted the comments on brain metastases:Furthermore, these investigations often do not consider many important nuances: for example, subjects with brain metastases are usually considered as an especial entity in tumor-specific trials, however this issue has not received a proper attention in the pan-cancer evaluation of targeted drugs [16,29,30,42,44,48,49,67,68,73,77,80,94,97,103,104,105,125,128].”

Reviewer 4 Report

Comments and Suggestions for Authors

Ijms-2900462

The authors summarized recent papers about the agnostic administration of targeted anticancer drugs, which is a promising method but has now some limitations. The authors arranged the recent major results of trials, well, then the manuscript has a good text for the new entrant in this academic and clinical field.

The reviewer will accept this manuscript if the authors respond some points below.

Major point

The manuscript has one conceptual figure (Figure 1). The potential readers who have recently entered this field may have difficulty understanding this image. What is the red circle on the bodies and what is the lines and cylinder in the central part? The authors should explain the figure in more detail in its Figure Legend.

Minor point

Some references may be registered in incorrect information. Ex) Ref 31 and 43 202 may be 2022 (please check all refs)

Author Response

Comment: The manuscript has one conceptual figure (Figure 1). The potential readers who have recently entered this field may have difficulty understanding this image. What is the red circle on the bodies and what is the lines and cylinder in the central part? The authors should explain the figure in more detail in its Figure Legend.

Response: We have extended explanations in the Figure legend.

Comment: Some references may be registered in incorrect information. Ex) Ref 31 and 43 202 may be 2022 (please check all refs)

Response: Thank you for noticing this! We have double-checked the references and corrected errors.

Round 2

Reviewer 4 Report

Comments and Suggestions for Authors

The authors extended the explanation of Fig 1. The reviewer additionally pointed out the following matter: at least, please write out the full form of TMB, MSI, and HRD in the Fig legend.